# Quantitative trait loci for grain mineral element accumulation in Vietnamese rice landraces

Hien Linh Tran[1,2]☯, Giang Thi Hoang[2,3]☯*, Nhung Thi Phuong Phung[2], Ham Huy Le[2,3], Alexandre Grondin[1]*, Pascal Gantet[1]

1 UMR DIADE, IRD, CIRAD, Université de Montpellier, Montpellier, France, 2 National Key Laboratory for Plant Cell Biotechnology, LMI RICE, Agricultural Genetics Institute, Hanoi, Vietnam, 3 VNU University of Engineering and Technology, VNU, Hanoi, Vietnam

☯ These authors contributed equally to this work.
* alexandre.grondin@ird.fr (AG); nuocngamos@yahoo.com (GTH)

**Data Availability Statement:** The data underlying the results presented in the study are available at https://doi.org/10.23708/A76RP9.

## Abstract

Rice (*Oryza sativa* L.) is a staple food for half of the world's population, and its biofortification is a key factor in fighting micronutrient malnutrition. However, harmful heavy metals tend to accumulate in rice grains due to soil and water contamination. Therefore, it is important to improve beneficial micronutrient contents and reduce the accumulation of undesirable metals in rice grain. To better characterize the genetic control of mineral accumulation in rice, we conducted association genetics on the ion contents of white and brown grains using a collection of 184 Vietnamese rice landraces. In total, 27 significant associations were identified and delimited into quantitative trait loci associated with macronutrients such as phosphorus, potassium or calcium; micronutrients such as iron or zinc; or toxic heavy metals such as arsenic and cadmium. Several genes related to ion homeostasis or ion transport were identified in the different quantitative trait loci. *LOC_Os10g30610*, present in qRAs10-1 associated with arsenic content in brown rice, encodes an ABC transporter (OsABCG25), which is involved in the silicon-induced formation of the Casparian strip in the rice exodermis and could act as a barrier restricting As diffusion within the root cortex. *LOC_Os05g04330*, present in qRP5-1 and associated with phosphorus content in brown rice, encodes a CHH methylation maintenance protein, and its expression is downregulated in roots in the presence of the *phosphorus uptake 1* (*Pup1*), suggesting a role for epigenetics in the regulation of phosphorus uptake and accumulation in grain. These findings reveal novel quantitative trait loci associated with grain ion content and candidate genes that are potentially valuable for breeding programs aimed at rice grain biofortification and reducing toxic metal accumulation.

## Introduction

Rice (*Oryza sativa* L.) is a staple food for more than half of the world's population, serving as a primary source of nutrients for billions of people worldwide. It is a significant contributor of

**Funding:** Hien Linh Tran is supported by the France Excellence Scholarship Program from the French Embassy in Vietnam. This work was supported by the Institut de Recherche pour le Développement (IRD), the "QUARION" EPPN European Plant Phenotyping Network (EPPN) 2020 project, the "RICE-VN" TRENPLIN-ASEAN prize awarded by the French Academy of Science and the French Ministry of Higher Education and Research (2023–2026). The funders had no role in study design, data collection and analysis, decision to publish, or preparation of the manuscript.

**Competing interests:** The authors have declared that no competing interests exist.

**Abbreviations:** BR, Brown rice; WR, White rice; qRAs1-1, QTL identifier (qR) for ion arsenic (qRAs) located on chromosome 1 (qRAs1) with specific identification number for this ion on chromosome 1 (qRAs1-1); ICP–MS, : Inductively coupled plasma mass spectrometry; BLINK, Bayesian-information and linkage-disequilibrium iteratively nested keyway.

calories while also supplying essential minerals such as calcium (Ca), phosphorus (P), and magnesium (Mg) and, to a lesser extent, copper (Cu), iron (Fe), manganese (Mn) and zinc (Zn) [1]. In some rice-consuming countries, the biofortification of Fe and Zn contents constitutes a crucial objective to reduce micronutrient malnutrition and subsequent effects on human health [2]. Fe deficiency impacts mostly young children and pregnant women in Southeast Asia, West Pacific nations and Africa, whereas Zn deficiency is more widespread in all developing countries [3]. Improvements in the Fe content to over 13 mg/kg and the Zn content to over 28 mg/kg in polished rice grain were set as international targets [4]. Although rice contains many beneficial minerals, it also carries the risk of containing harmful metals for human health, such as arsenic (As) and cadmium (Cd). Rice plants grown in contaminated As and Cd soils are affected in their development, yield and other desired market traits such as the head rice recovery percentage, or consumer traits such as the cooking characteristics [5–7]. Furthermore, accumulations of As and Cd in rice are often associated with reductions in other beneficial ions in grains, as these harmful ions can reduce the accessibility of beneficial ions in the soil or share similar uptake and translocation mechanisms. For instance, As can bind with Fe and Mn oxides and competes with silicon (Si) for uptake [8]. Similarly, NATURAL RESISTANCE-ASSOCIATED MACROPHAGE PROTEIN 5 (OsNRAMP5) acts as a major transporter of Cd and Mn in the roots [9, 10] while OsNRAMP1 is another metal transporter that transports both Cd and Fe [11]. Absorption of Cd can also be mediated by IRON-REGULATED TRANSPORTERS 1 and 2 (OsIRT1 and OsIRT2) [12]. Therefore, when enhancing rice biofortification, caution should be taken to avoid trade-offs caused by the accumulation of harmful elements.

Better agronomical practices combined with varietal improvement successfully improved Zn content by over 24 mg/kg in some released biofortified rice in Bangladesh [13]. The limited variation in the contents of some ions, such as Fe, in polished rice impedes biofortification through conventional breeding. The complex genetic and metabolic networks controlling the ion content in grain, which is highly influenced by the environment, represent another limitation [2, 14]. Transgenic approaches represent an alternative to circumvent these constraints, as constitutive expression of genes involved in ion uptake, transport, storage or metabolism may be less sensitive to the environment and surpass natural variation. For example, the overexpression of *nicotianamine synthase 2* (*NAS2*), which catalyzes the synthesis of the divalent metal chelator nicotianamine acid, resulted in increases of up to 19 mg/kg in Fe and 76 mg/kg in Zn in rice endosperm [15]. However, policy regulations on transgenics limit the use of this approach, and it is important to continue exploring the diversity in ion content in rice grain to identify potential donor parents and quantitative trait loci (QTLs) that could be deployed in breeding. The uncoupling of grain biofortification for beneficial nutrients from the accumulation of harmful metals remains possible, especially through the identification of uptake and transport mechanisms specifically related to these harmful metals [16]. Studies have shown that natural variability in As accumulation in rice varieties exists, with varieties accumulating significantly lower amounts of As than others (up to 20–30 times less) [17–19]. Variation in grain Cd accumulation in a rice mapping population allowed the identification of the *qGCd7* QTL, in which the *HEAVY METAL ATPASE 3* (*OsHMA3*) gene is responsible for limiting root-to-shoot translocation of Cd by its selective sequestration within the vacuole [20]. A better understanding of the genetic and physiological determinants of beneficial and harmful ion accumulation in grains is needed to more efficiently improve rice grain nutritional quality.

Biparental populations (for example, from two contrasted parents or multiparent advanced generation intercross population) can target the identification of genetic determinants of only some ions that present contrasted accumulation in parents [21, 22]. More recently, the capacity to genotype rice diversity collections has been useful for studying the diversity in the grain

ionome and identifying associated QTLs and genes through genome-wide association studies (GWAS) [23, 24]. Studies on "Rice Diversity Panel 1", a collection of accessions representative of *O. sativa* global diversity, identified 330 significant SNPs associated with As, Cu, molybdenum (Mo), and Zn accumulation in grain [25, 26]. By performing a GWAS in a different collection, Yang et al. [27] identified 72 QTLs associated with ion variation and provided evidence for the causal roles of three genes, the sodium transporter gene *HIGH AFFINITY K + TRANSPORTER 1;5 (OsHKT1;5)* for sodium (Na), *MOLYBDATE TRANSPORTER 1;1 (OsMOT1;1)* for Mo and *GRAIN NUMBER, PLANT HEIGHT, AND HEADING DATE7 (Ghd7)* for nitrogen (N). Another study using an *O. sativa* spp. indica panel identified the *NICOTIANAMINE SYNTHASE 3 (OsNAS3)* gene as potentially responsible for the effect of a quantitative trait locus (QTL) associated with Zn content in rice grain [28]. This highlights the usefulness of rice panels to reveal complementary QTLs and genes involved in rice grain ion content. However, most of these association studies were conducted separately for brown rice (BR) and white rice (WR), and only a few studies have compared the genetic basis of both brown and white grains collected from the same set of rice plants [29]. Research on QTLs controlling BR and WR grain ion contents measured in the same plants is becoming increasingly important as brown rice consumption rises.

Vietnamese rice landraces are adapted to a range of environments from North to South Vietnam and present an interesting genetic diversity that has been overlooked. One study using 94 black Vietnamese rice accessions identified original associations for the levels of anthocyanin and flavonoids present in whole grains [30]. Moreover, a genotyped collection of Vietnamese rice landraces originating from various agrosystems across Vietnam was previously established and sequenced [31]. A GWAS of this collection revealed original QTLs associated with root traits [32], panicle development [33], leaf biomass [34], drought resistance [35], jasmonate regulation [36], salt resistance [37] and phosphorous deficiency [38]. However, the diversity in grain ion content is unknown in the Vietnamese rice landraces collection. In this study, we aimed at characterizing the genetic diversity for the contents of 23 ions in brown and white rice grains from plants in this Vietnamese rice landrace collection grown in a field in Vietnam. We further performed a GWAS to identify QTLs governing the ion contents in brown and white rice grains.

## Materials and methods

### Plant materials and genotypic data

In this study, 184 rice genotypes were used, including 181 Vietnamese landraces (113 indica, 62 japonica and 6 admixed accessions) [31] and 3 reference genotypes (Nipponbare, Azucena and IR64; S1 Table). These accessions were collected and are available at the Plant Resources Center (PRC, An Khanh commune, Hoai Duc district, Hanoi city, Vietnam), which acts for the Vietnamese government to preserve the seeds of traditional varieties.

### Field experiment

The field experiment was performed in Hai Phong, Vietnam (20° 51' 59" N and 106° 40' 57" E), between July and November 2019. It was conducted in a field normally used for rice cultivation, therefore no special authorization was required. Seeds were sown in seedling beds for two weeks, and the seedlings were further transplanted into an open field under irrigated lowland conditions (S1 Fig). The rice accessions were grouped according to their flowering time: 67 accessions in the early-flowering group (less than 85 days), 72 accessions in the medium-flowering group (85–119 days), 23 accessions in the late-flowering group (120–154 days), and 22 accessions in the very late-flowering group (more than 155 days). The four flowering groups

were divided into blocks (12 blocks in early, 12 blocks in medium, 4 blocks in late, and 4 blocks in very late) with 50 cm spacing between each pair of blocks and 75 cm spacing between flowering groups. Each accession was planted in three plots within their flowering group, which corresponded to replicates. Each plot was 1 m$^2$ in area and contained 25 plants separated by 25 cm between and within rows. The grains from all 25 plants per plot were harvested when the plants reached maturity and were air-dried and stored at 4˚C. After drying, the inedible outer hulls of the grains were manually removed to obtain brown rice (BR). White rice (WR), corresponding to the grains without the hull and the bran (including the aleurone layer and the cereal germ), was obtained using a huller machine. Ten BR and WR grains from each plot were finely ground separately using a nonmetal grinder (HARIO Ceramic Coffee Mill Slim Plus Grinder MSS-1DTB) before ion content analysis.

## Grain ion phenotyping

The ion contents were measured at the ionomic platform of the University of Nottingham by inductively coupled plasma–mass spectrometry (ICP–MS) following a similar procedure to [39]. Briefly, ground grains were placed in Pyrex test tubes, and their weights were measured. The tubes were complemented with 1 mL of trace metal grade nitric acid Primar Plus (spiked with an indium internal standard). The samples were subsequently subjected to a preliminary digestion process at room temperature in a fume hood for 20 hours. The samples were transferred to dry block heaters and digested at 115˚C for 4 hours. After the samples had cooled to room temperature, 1 mL of hydrogen peroxide was introduced into the tubes, and the samples were subjected to an additional 2-hour digestion in a dry block heater at 115˚C. The resulting digested samples were then diluted to a final volume of 10 mL with 18.2 MΩcm Milli-Q Direct water. ICP–MS was performed using a PerkinElmer NexION 2000 in collision mode with Helium (He). Twenty-three elements, including lithium (Li), boron (B), sodium (Na), magnesium (Mg), phosphorus (P), sulfur (S), potassium (K), calcium (Ca), manganese (Mn), iron (Fe), cobalt (Co), nickel (Ni), copper (Cu), zinc (Zn), arsenic (As), selenium (Se), rubidium (Rb), strontium (Sr), molybdate (Mo), cadmium (Cd), lead (Pb), titanium (Ti) and chromium (Cr), were analyzed in both brown and white rice samples. A liquid reference material composed of pooled samples of digested grain powders was prepared before the beginning of the sample run and was used throughout the whole run. It was run after every ninth sample in all the ICP–MS sample sets to correct for variation between and within the ICP–MS analysis runs. Single-element standard solutions were used for calibration. Sample concentrations were calculated using an external calibration method in the instrument's software.

## Statistical analysis

The mean and standard deviation of three repetitions as well as the coefficient of variation for each genotype were measured for all the traits using RStudio version 2022.07.1. A two-way ANOVA (using the aov function) was performed to test the effects of replication and genotype. Broad-sense heritability (H$^2$) was calculated as H$^2$ = ($F$ value − 1)/$F$ value, where the $F$ value for the genotype effect was extracted from the ANOVA as described by [34]. Correlations between all traits were analyzed using Pearson's correlation test with a confidence level of 0.95. These correlations were visualized using the corrplot function from the R corrplot package. To perform GWAS, BLUEs per genotype for each ion in BR and WR were calculated in the StatgenSTA package [40] using the spatial analysis of field trials with splines (SpATS) function [41] and considering a resolvable incomplete block design according to the following model:

$$\text{trait} = \text{repID} + \text{repID} : \text{subBlock} + \text{genotype} + \varepsilon$$

where repetition (repID) is considered a fixed effect and repID:subBlock is considered a random effect.

## Genome-wide association studies

GWAS was performed with the GAPIT3 package using a previously published genotypic dataset obtained from 184 accessions from the Vietnamese rice collection, which consisted of 21623 SNP markers [31]. The structure of this population was previously analyzed, and six genotypic groups were identified [31]. Associations between genotypes and phenotypes (BLUEs) were performed using different models: the mixed linear model (MLM), generalized linear model (GLM), multilocus mixed model (MLMM), conditional masked language modeling (CMLM), enriched compressed mixed linear model (ECMLM) and Bayesian-information and linkage-disequilibrium iteratively nested keyway (BLINK). Among these methods, BLINK is considered to have the highest statistical power and computing efficiency [42]. Therefore, BLINK was chosen to filter significant SNPs according to the Bonferroni threshold with a significance level of 0.05. Quantile–quantile plots (QQ plots) and Manhattan plots were generated using GAPIT3 [42].

## Linkage disequilibrium heatmap and identification of candidate genes

To delimit QTL regions, pairwise linkage disequilibrium was calculated between significant SNPs and the surrounding SNPs using the LDheatmap package in R [43]. The QTL intervals were restricted to a region where the squared allele frequency correlation ($r^2$ values) between markers exceeded 0.4. If the previously delimited region was smaller than 50 kb, the QTL region was extended to include a 50 kb range both upstream and downstream of the significant markers. Genes within the QTL were identified using the MSU Rice Genome Annotation Project Release 7 (http://rice.uga.edu/) and RAPDB version IRGSP-1.0 2023-09-07 (https://rapdb.dna.affrc.go.jp/). The candidate genes in each QTL region were subsequently selected based on potential links between their predicted function and the ion of interest. The expression levels of the selected candidate genes were retrieved from RiceXPRo version 3.0 (https://ricexpro.dna.affrc.go.jp/index.html) [44].

## Results

### Variation in the grain ion contents of Vietnamese rice landraces

To analyze the variability in ion accumulation in BR and WR in the Vietnamese rice collection, the contents of 23 different ions were analyzed in grains collected from a field experiment performed in Vietnam. ICP–MS analyses revealed the presence of Mg, P, S, K, Ca, Mn, Fe, Co, Ni, Cu, Zn, As, Rb, Sr, Mo and Cd, whereas the Li, B, Na, Ti, Cr, Se and Pb concentrations were below the quantification limits. These latter elements were consequently excluded from further analysis. Most ions presented higher values in BR than in WR, except for Mo and Cd (S2 Fig). In both BR and WR, macronutrients (P, K, Mg, S and Ca) were predominant, as expected, with P having the highest concentration in BR (3369 ppm) and S having the highest concentration in WR (1255 ppm). Among the micronutrients, Co had the lowest concentrations in both BR and WR (0.03 ppm and 0.02 ppm, respectively). In BR, K, Mg, P, and S presented the smallest variations within the collection, with coefficients of variation varying between 11% and 12%. In contrast, Co, As, Mo, and Cd showed the greatest variations, with the coefficients of variation ranging from 33% to 47% in both BR and WR. Significant genotypic effects were observed for all ions with high values of broad-sense heritability ($\geq 0.71$ for BR and $\geq 0.73$ for WR; Table 1), indicating that the grain ion content is genetically controlled in the Vietnamese rice collection.

**Table 1. Phenotypic variation and broad sense heritability of grain ion content in brown and white rice in the Vietnamese rice collection.**

| Grain type | Ion | Mean (ppm) | SD | Min (ppm) | Max (ppm) | CV (%) | Replication effect (p-value) | Genotype effect (p-value) | H² |
|---|---|---|---|---|---|---|---|---|---|
| Brown rice | P | 3369.13 | 383.26 | 2343.73 | 4956.05 | 11.38 | 0.4430 | <0.001 | 0.79 |
| | K | 2653.99 | 321.99 | 1908.69 | 3788.00 | 12.13 | 0.0298 | <0.001 | 0.86 |
| | Mg | 1357.14 | 158.92 | 916.16 | 1927.55 | 11.71 | <0.001 | <0.001 | 0.82 |
| | S | 1310.55 | 166.01 | 847.77 | 1937.29 | 12.67 | <0.001 | <0.001 | 0.73 |
| | Ca | 101.47 | 19.37 | 54.19 | 169.55 | 19.09 | 0.8235 | <0.001 | 0.71 |
| | Zn | 35.10 | 5.76 | 20.76 | 64.50 | 16.41 | <0.001 | <0.001 | 0.85 |
| | Rb | 27.45 | 5.82 | 13.30 | 49.93 | 21.20 | 0.0323 | <0.001 | 0.84 |
| | Mn | 23.98 | 4.25 | 11.51 | 35.68 | 17.72 | <0.001 | <0.001 | 0.84 |
| | Fe | 10.39 | 2.04 | 6.63 | 21.59 | 19.63 | <0.001 | <0.001 | 0.74 |
| | Cu | 4.02 | 0.78 | 2.02 | 6.80 | 19.40 | <0.001 | <0.001 | 0.90 |
| | Mo | 1.02 | 0.44 | 0.24 | 3.17 | 43.14 | <0.001 | <0.001 | 0.98 |
| | Ni | 0.95 | 0.28 | 0.24 | 1.89 | 29.47 | <0.001 | <0.001 | 0.90 |
| | Sr | 0.27 | 0.08 | 0.10 | 0.69 | 29.63 | 0.0151 | <0.001 | 0.91 |
| | Cd | 0.21 | 0.10 | 0.02 | 0.53 | 47.62 | 0.0011 | <0.001 | 0.95 |
| | As | 0.16 | 0.07 | 0.01 | 0.35 | 43.75 | 0.1528 | <0.001 | 0.78 |
| | Co | 0.03 | 0.01 | 0.01 | 0.10 | 33.33 | 0.2419 | <0.001 | 0.88 |
| White rice | S | 1254.91 | 168.63 | 766.00 | 1958.37 | 13.44 | <0.001 | <0.001 | 0.75 |
| | P | 1246.63 | 260.15 | 540.82 | 2340.61 | 20.87 | <0.001 | <0.001 | 0.92 |
| | K | 982.53 | 237.79 | 499.89 | 1944.03 | 24.20 | 0.0836 | <0.001 | 0.94 |
| | Mg | 377.94 | 109.77 | 148.12 | 865.10 | 29.04 | 0.0779 | <0.001 | 0.92 |
| | Ca | 54.11 | 13.07 | 20.30 | 101.37 | 24.15 | 0.0112 | <0.001 | 0.80 |
| | Zn | 28.47 | 5.31 | 16.94 | 47.70 | 18.65 | <0.001 | <0.001 | 0.91 |
| | Mn | 10.77 | 2.09 | 6.23 | 18.72 | 19.41 | 0.0059 | <0.001 | 0.92 |
| | Rb | 9.70 | 2.42 | 4.59 | 20.69 | 24.95 | 0.0323 | <0.001 | 0.89 |
| | Fe | 4.04 | 1.56 | 1.18 | 11.30 | 38.61 | <0.001 | <0.001 | 0.73 |
| | Cu | 3.39 | 0.64 | 1.61 | 5.92 | 18.88 | <0.001 | <0.001 | 0.94 |
| | Mo | 1.01 | 0.46 | 0.23 | 3.25 | 45.54 | <0.001 | <0.001 | 0.98 |
| | Ni | 0.63 | 0.22 | 0.18 | 1.59 | 34.92 | <0.001 | <0.001 | 0.90 |
| | Cd | 0.21 | 0.10 | 0.01 | 0.53 | 47.62 | <0.001 | <0.001 | 0.95 |
| | As | 0.11 | 0.05 | 0.01 | 0.29 | 45.45 | 0.0495 | <0.001 | 0.82 |
| | Sr | 0.09 | 0.03 | 0.03 | 0.38 | 33.33 | 0.0951 | <0.001 | 0.89 |
| | Co | 0.02 | 0.01 | 0 | 0.05 | 50.00 | <0.001 | <0.001 | 0.90 |

SD: standard deviation; Min: minimum mean value per genotype observed in the collection; Max: maximum mean value per genotype observed in the collection; CV: coefficient variation; H²: broad sense heritability. p-values from two-way ANOVA tests are represented.

To check for correlations between ions, Pearson's correlation analysis was performed (Fig 1). Moderate to very strong positive correlations were observed between the P, K and Mg contents for both BR and WR (R = 0.54 to R = 0.87 between combinations in BR and WR). In BR, moderate positive correlations were observed between the S and P contents (R = 0.51) and between the Sr and As contents (R = 0.54). In WR, Mn was moderately positively correlated with Mg, K and Sr (R > 0.50), Rb was also moderately positively correlated with Mg and K (R > 0.55), and Mo was moderately positively correlated with Zn (R = 0.50). For both BR and WR, significant negative correlations were observed between the Ni and As contents (R = −0.34 and R = −0.36, respectively), between the Ni and Mo contents (R = −0.46 and R = −0.39, respectively) and between the Cd and Mo contents (R = −0.45 and R = −0.47, respectively).

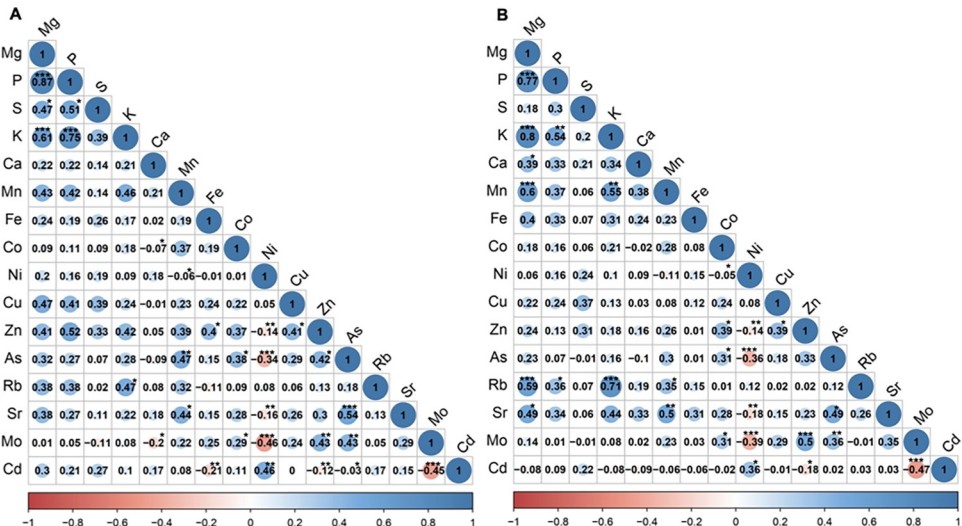

**Fig 1.** Correlation between ion content in brown rice (A) and white rice (B) grains. Color gradients indicate the Pearson's correlation coefficient. Non-significant correlations at a p-value threshold of 0.05 are indicated with a cross.

These results indicate strong associations for the ion content in BR and WR (e.g., P, K and Mg).

## Identification of QTLs associated with grain ion content in Vietnamese rice landraces

The high diversity and high heritability observed in the Vietnamese rice collection for ion content in both BR and WR suggest that these traits are amenable for association genetic analysis. To identify genomic regions associated with grain ion content in Vietnamese rice landraces, we carried out a GWAS on the 16 detected ions in BR and WR rice using a previously described genotypic dataset composed of 21623 SNP markers [31]. We identified a total of 27 associations considering the Bonferroni threshold (Table 2 and S3 Fig). Among them, 14 associations passed the significance threshold in BLINK and other GWAS models, and 13 exclusively passed the significance threshold in the BLINK model. Significant associations on chromosome 7 for Cu and on chromosome 10 for As were observed in both BR and WR. Additionally, a similar association was observed on chromosome 11 for both Cu and Fe in BR (Table 2).

Pairwise linkage disequilibrium was calculated between significant SNPs to define the size of each association, which was further referred to as a QTL (Table 2). We investigated whether the QTLs identified in our study colocalized with previously identified QTLs for root and panicle architecture and development, and salt and drought stress tolerance-related traits in the same Vietnamese rice collection [32–35, 37]. We found that qRMo1-1 overlaps with QTL q48, which is related to the number of crown roots per tiller. qRAs1-1 and qRCa4-1 also overlap with QTL_2 and QTL_11, which are related to the ratio of chlorophyll a to chlorophyll b in leaves [37]. qRK10-2 and qRCa4-1 overlap with a QTL associated with leaf mass and yield-related traits [33, 34]. Finally, qRCd11-1 overlaps with QTLs associated with drought tolerance-related traits [35]. In addition, we looked for colocalization between QTLs for grain ion content identified in the Vietnamese rice collection and QTLs for grain ion content identified in other rice populations in the literature. On chromosome 1, qRAs1-1 overlaps with qAs1-3, a QTL for grain As content identified in a rice diversity panel composed of 202 accessions from

**Table 2. List and position of QTLs identified in the Vietnamese collection for ion content in brown rice (BR) and white rice (WR) grains.**

| Ion | QTL | Grain type | Chr | SNP | SNP position | QTL position |
|---|---|---|---|---|---|---|
| As | qRAs1-1 | BR | 1 | Sj01_33174934R | 33174934 | 33060660–33174934 |
| As | qRAs4-1 | WR | 4 | **Dj04_03138652R** | 3138652 | 2977218–3143723 |
| As | qRAs4-2 | WR | 4 | Dj04_32371574F | 32371574 | 32347540–32371574 |
| As | qRAs10-1 | BR/WR | 10 | **Sj10_15968777F** | 15968777 | 15873781–16018764 |
| As | qRAs11-1 | BR | 11 | **Dj11_05821648F** | 5821648 | 5747401–5875727 |
| Ca | qRCa3-1 | BR | 3 | **Dj03_29401407R** | 29401407 | 29303078–29439457 |
| Ca | qRCa4-1 | BR | 4 | **Sj04_04361919F** | 4361919 | 4307480–4399327 |
| Ca | qRCa6-1 | BR | 6 | **Sj06_24039978F** | 24039978 | 23989978–24089978 |
| Ca | qRCa9-1 | BR | 9 | **Dj09_15688119R** | 15688119 | 15638119–15738119 |
| Cd | qRCd10-1 | BR | 10 | Dj10_02130045F | 2130045 | 2081167–2151377 |
| Cd | qRCd11-1 | BR | 11 | Dj11_06632630F | 6632630 | 6582630–6682630 |
| Cu | qRCu7-1 | BR/WR | 7 | **Sj07_25260997F** | 25260997 | 25232425–25360187 |
| Cu/Fe | qRCuFe11-1 | BR | 11 | Dj11_20389966F | 20389966 | 20336015–20407398 |
| Fe | qRFe3-1 | WR | 3 | **Dj03_28302202R** | 28302202 | 28109488–28355049 |
| Fe | qRFe10-1 | WR | 10 | Sj10_19727114F | 19727114 | 19727114–19788660 |
| K | qRK5-1 | BR | 5 | Dj05_03417694F | 3417694 | 3365689–3504647 |
| K | qRK5-2 | BR | 5 | Dj05_05422273F | 5422273 | 5390697–5451215 |
| K | qRK10-1 | BR | 10 | Dj10_04097517F | 4097517 | 4046874–4097937 |
| K | qRK10-2 | BR | 10 | **Sj10_18059798F** | 18059798 | 17948827–18064123 |
| Mn | qRMn4-1 | BR | 4 | Sj04_25552228F | 25552228 | 25398103–25552228 |
| Mo | qRMo1-1 | WR | 1 | **Sj01_28315371F** | 28315371 | 27868310–28390748 |
| P | qRP5-1 | BR | 5 | **Dj05_01901555F** | 1901555 | 1884166–1970447 |
| S | qRS7-1 | BR | 7 | Dj07_21988531R | 21988531 | 21896842–21988531 |
| S | qRS12-1 | BR | 12 | Dj12_6190592R | 6190592 | 6136693–6190592 |
| Sr | qRSr3-1 | WR | 3 | **Sj03_14341377R** | 14341377 | 14266371–14341377 |
| Sr | qRSr8-1 | WR | 8 | Sj08_18311304F | 18311304 | 18261304–18361304 |
| Zn | qRZn1-1 | WR | 1 | **Sj01_03956268F** | 3956268 | 3956268–4153223 |

Chr: Chromosome; BR: brown rice; WR: white rice. SNP in bold font passed the significance threshold in BLINK and other GWAS models. SNP in regular font passed the significance threshold in the BLINK model only.

**Table 3. List of candidate genes identified in QTLs associated with P, K and As content in brown rice (BR) and white rice (WR) grains.**

| QTL | Ion | Chr | Grain type | Gene ID | Gene annotation - hypothetical function | Reference |
|---|---|---|---|---|---|---|
| qRP5-1 | P | 5 | BR | LOC_Os05g04330/ Os05g0133900 | DNA methyltransferase protein, DRM3/possibly regulated by *Pup1* in root under P starvation | [47] |
| qRK5-1 | K | 5 | BR | LOC_Os05g06660/ Os05g0158500 | Putative Serine Carboxypeptidase homologue, OsSCP26/upregulated under K starvation | [48] |
| qRAs10-1 | As | 10 | BR—WR | LOC_Os10g30610/ Os10g0442900 | ABC transporter, OsABCG25/involved in casparian strip formation mediated by Si | [49] |
| qRAs11-1 | As | 11 | BR | LOC_Os11g10510/ Os11g0210500 | alcohol dehydrogenase, *OsADH2*/indirectly regulates As uptake by roots | [52] |

Chr: Chromosome.

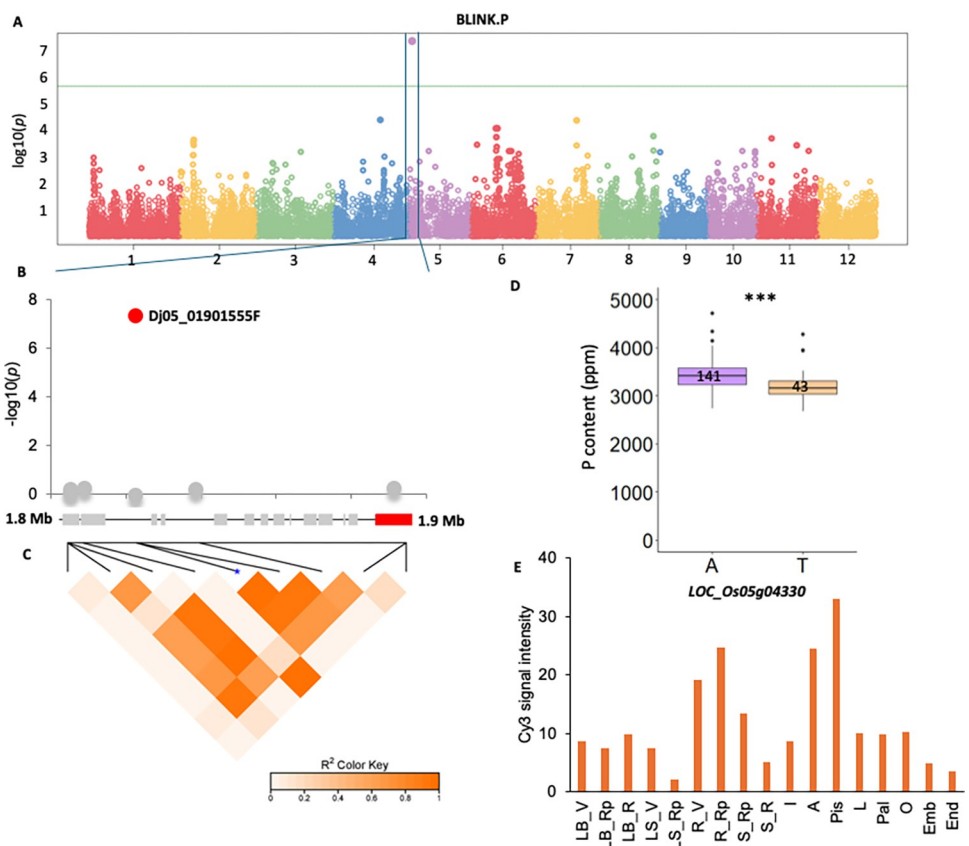

**Fig 2. Genome-wide association mapping of P content in brown rice grain measured in the Vietnamese rice collection. A**. Manhattan plot representing the p-values of associations between SNPs and grain P content using the BLINK model in GAPIT. The solid green line represents the Bonferroni threshold. **B**. Focus on the association between SNP Dj05_01901555F and grain P content on chromosome 5. **C**. Definition of QTL qRP5-1 based on blocks of linkage disequilibrium around Dj05_01901555F. The genes present in the QTL region are represented by grey rectangles. The selected candidate gene *LOC_Os05g04330* is highlighted in red. **D**. Box plots representing the polymorphism distribution at SNP Dj05_01901555F in the Vietnamese collection. The number of genotypes in each haplotype group is represented. **E**. Expression levels of *LOC_Os05g04330* at different developmental stages. Expression data in different plant tissues were retrieved from the Rice Xpro database [44] as follow; LB_V: Leaf blade at vegetative stage; LB_Rp: Leaf blade at reproductive stage; LB_R: Leaf blade at ripening stage; LS_V: Leaf sheath at vegetative stage; LS_Rp: Leaf sheath at reproductive stage; R_V: Root at vegetative stage; R_Rp: Root at reproductive stage; S_Rp: Stem at reproductive stage; S_R: Stem at ripening stage; I: Inflorescence of 3.0–4.0 mm; A: Anther of 0.7–1.0 mm; Pis: Pistil of 10–14 cm panicle; L: Lemma of 4.0–5.0 mm floret; Pal: Palea of 4.0–5.0 mm floret; O: Ovary 3 days after flowering; Emb: Embryo at 14 days after flowering; End: Endosperm at 14 days after flowering.

the USDA-ARS Rice MiniCore collection [45]. Furthermore, on chromosome 11, qRCd11-1 (6582630–6682630 bp) is positioned close to qCd11-1 (6233769–6354200 bp), a QTL associated with grain Cd content identified in a subset of 698 varieties present in the 3K rice genome project [46].

## Potential candidate genes related to grain ion content in Vietnamese rice landraces

Genes present in each QTL were investigated (S2 Table) for their putative functions in determining the grain ion content based on their functional annotations and a literature survey. Five potential candidate genes were selected (Table 3). In qRP5-1 associated with grain P content, *LOC_Os05g04330*, which encodes a DNA CHH methylation maintenance protein, is

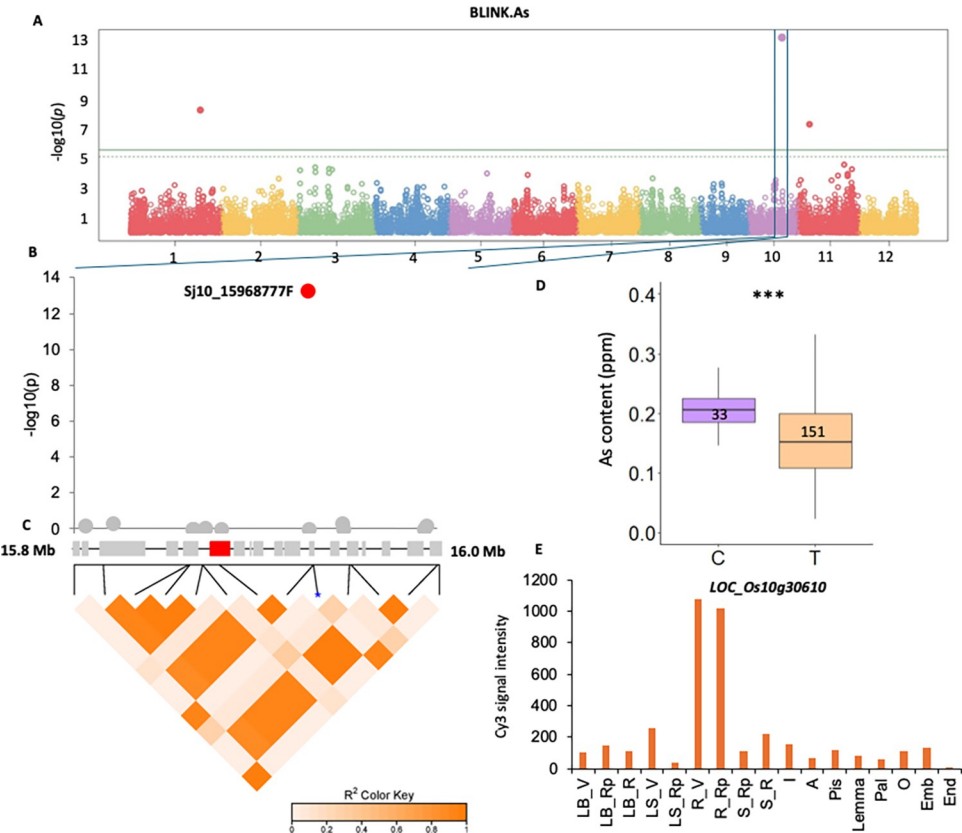

**Fig 3. Genome-wide association mapping of As content in brown and white rice grains measured in the Vietnamese rice collection. A**. Manhattan plot representing the p-values of associations between SNPs and grain As content using the BLINK model in GAPIT. The solid green line represents the Bonferroni threshold and the dotted green line represents the FDR threshold. **B**. Focus on the association between SNP Sj10_15968777F and grain As content on chromosome 10. **C**. Definition of QTL qRAs10-1 based on blocks of linkage disequilibrium around Sj10_15968777F. The genes present in the QTL region are represented by grey rectangles. The selected candidate gene *LOC_Os10g30610* is highlighted in red. **D**. Box plots representing the polymorphism distribution at SNP Sj10_15968777F in the Vietnamese collection. The number of genotypes in each haplotype group is represented. **E**. Expression levels of *LOC_Os10g30610* at different developmental stages. Expression data in different plant tissues were retrieved from the Rice Xpro database [44] as follow; LB_V: Leaf blade at vegetative stage; LB_Rp: Leaf blade at reproductive stage; LB_R: Leaf blade at ripening stage; LS_V: Leaf sheath at vegetative stage; LS_Rp: Leaf sheath at reproductive stage; R_V: Root at vegetative stage; R_Rp: Root at reproductive stage; S_Rp: Stem at reproductive stage; S_R: Stem at ripening stage; I: Inflorescence of 3.0–4.0 mm; A: Anther of 0.7–1.0 mm; Pis: Pistil of 10–14 cm panicle; L: Lemma of 4.0–5.0 mm floret; Pal: Palea of 4.0–5.0 mm floret; O: Ovary 3 days after flowering; Emb: Embryo at 14 days after flowering; End: Endosperm at 14 days after flowering.

located 6404 bp downstream of the significant Dj05_01901555F SNP marker (Fig 2A–2C). Haplotype analysis of this SNP revealed that the T allele present in a minority of landraces was associated with increased grain P content (3.6% increase on average; Fig 2D). *LOC_Os05g04330* is highly expressed in roots during the vegetative stage as well as in anthers and pistils under optimal conditions [44] (Fig 2E). Under P-starvation stress, downregulation of this gene has been observed in a near-isogenic line harboring the *phosphorus uptake 1* (*Pup1)* QTL [47]. *LOC_Os05g06660*, located in qRK5-1, also appeared interesting, as it encodes a serine carboxypeptidase homologous to OsSCL26 (Table 3 and S4 Fig). OsSCL26 plays an important role in the homeostasis of P in rice [48], an ion that was highly correlated with K in our study.

Several genes were also linked to the grain As content in qRAs10-1 and qRAs11-1. In qRA10-1, *LOC_Os10g30610*, located 39981 bp upstream of the significant SNP marker Sj10_15968777F, encodes an ABC transporter (OsABCG25), which is involved in the silicon-induced formation of the Casparian strip in the rice exodermis (Fig 3A–3C and Table 3) [49]. Haplotype analysis revealed that allele T is present in the majority of Vietnamese rice landraces and provides a lower grain As content than the C allele (24% reduction on average; Fig 3D). *OsABCG25* is expressed mainly in the roots (Fig 3E). This gene is upregulated under high-As treatment and may be involved in the homeostasis of As in rice [50]. In qRAs11-1, *LOC_Os11g10510* encodes ALCOHOL DEHYDROGENASE 2 (OsADH2), which regulates silicate transporters also acting as As transporters in rice (S5 Fig and Table 3) [51, 52]. Overall, some of these genes represent new potential candidates for P uptake and exclusion of heavy metals such as As in rice plants and ultimately grains.

## Discussion

Improving the quality of rice grain requires improving their mineral content. However, while some minerals are important for human health, others can be toxic. Many efforts have focused on the biofortification of rice grain with Fe and Zn, but further increases in these nutrients in rice grain are needed to fight malnutrition worldwide. Therefore, it is important to characterize the diversity of ion contents in rice grain, identify the genetic determinants controlling grain ion accumulation and determine the physiological significance of these mechanisms to avoid trade-offs related to the accumulation of toxic heavy metals. To contribute to this objective, we assessed the contents of 16 different ions in BR and WR grains within a collection of 184 sequenced Vietnamese rice landraces and observed a large diversity for most ions. A GWAS identified 27 QTLs associated with the amounts of important macronutrients, such as P or K; micronutrients, such as Fe or Zn; or toxic elements, such as As and Cd. These QTLs contain relevant candidate genes related to ion transport or ion homeostasis adjustment in rice.

In the Vietnamese rice collection, P, K, Mg and S were the most abundant ions in both BR and WR grains, whereas Cd, As, Sr and Co were among the least abundant. The ion contents observed in our study were of the same order of magnitude as those reported in previous rice grain ionome studies [24, 28, 53]. Interestingly, although most ions were more concentrated in BR than in WR, no significant differences in ion content were observed between the two types of rice for Mo and Cd. This was previously observed for Cd [54, 55] and suggests that Cd, and most likely Mo, is not significantly lost during the milling process because of its accumulation in the endosperm rather than in the outer husk and bran layers [55]. The variations in the grain ion contents in the BR and WR samples also aligned with findings from previous studies [24, 28], where K, Mg, and P had the smallest variations, whereas Fe, Zn, As, Mo, and Cd had the largest variations. In our study, Fe and Zn varied from 6.63 to 21.59 ppm and from 20.76 to 64.50 ppm, respectively, in the BR grains. These values are higher than the maximum Fe (16.58 ppm) content observed in BR grains under flooded conditions in the USDA Core Collection by [24]. This indicates that the QTLs identified in our study constitute a good resource for rice grain biofortification toward the international targets (13 to 15 ppm for Fe and 28 ppm for Zn, considering rice daily consumption in countries where rice is a staple food) [4]. Vietnamese landraces with high grain Fe contents may therefore represent interesting donors for breeding programs aimed at rice grain Fe biofortification [56, 57]. Given that grain ion content is influenced by environmental factors whose frequencies are increasing due to climate change [58], it is important to develop climate-resilient rice varieties while ensuring the maintenance of grain quality.

High positive correlations were observed in both BR and WR between the chemical analogs K and Rb, which are known to share similar uptake mechanisms in plants [59]. Ca and Sr, which are also chemical analogs, were not correlated in either BR or WR in our study, possibly because of the low Sr content in the soil. However, high correlations were observed between P, K, and Mg in BR grains, although these ions do not share similar uptake mechanisms and are not tightly chemically coupled in the soil solution [60]. High correlations between P, K, and Mg were previously reported for BR in a set of widely diverse rice germplasms [24] as well as in other species, such as *Arabidopsis thaliana* [61] and maize [62]. The link between these ions in rice grains may be driven by phytate levels, as most of the P in rice grain is found in the form of mixed K–Mg salts of phytic acid in the aleurone layer and germ [24, 63]. Further evidence confirming the robustness of this correlation came from the colocalization of QTLs for P, K and Mg contents in BR grains in a biparental population [64]. Interestingly, we observed that the correlations between these three ions remained present in WR grains, indicating that other mechanisms independent of phytates within the aleurone layer coregulate P, K and Mg contents in the endosperm. Moreover, significant negative correlations between As and Ni contents in BR and WR grains were observed in our study, as previously described by Williams et al. [65]. Significant negative correlations were also observed between Mo and Cd in both BR and WR grains. In line with these results, it has been reported that Mo alleviates Cd uptake, translocation and toxicity in rice [66] and other species, such as *Brassica napus* [67]. The mechanisms involved in this response remain elusive but may be related to the beneficial physiological effects of Mo, which balance Cd stress rather than competition for uptake [66, 67]. Overall, our results support previous observations while providing further indications of the links between ion accumulation within different rice grain tissues, i.e., the aleurone layer versus the endosperm.

The GWAS revealed 27 significant SNPs associated with BR and WR grain ion contents. We extended these associations to QTL regions using linkage disequilibrium as a criterion to group SNPs together. The QTL qRAs1-1 identified in this study for As overlapped with the QTL qAs1-3, which was previously associated with the same traits but was identified in different rice populations [45]. Furthermore, the QTL qRCd11-1 is located close to another QTL associated with Cd in another rice population [46]. This underscores the robustness of these two QTLs and additionally validates our association study to some extent. Interestingly, five QTLs identified in this study (qRMo1-1, qRAs11-1, qRCa4-1, qRK10-2 and qRCd11-1) were found to overlap with QTLs for morphological and functional traits identified in prior studies using the same rice collection. A QTL for Mo (qRMo1-1) was found to overlap with q48 and was associated with the number of crown roots per tiller [32]. Mo uptake occurs predominantly in roots via MO TRANSPORTER 1 (MOT1) [68] and contributes to the structuration and activity of enzymes such as nitrate reductase [69]. The colocalization of qRMo1-1 and q48 suggests that the number of crown roots is important for the accumulation of Mo and ultimately its translocation to grain, likely via better Mo uptake capacity through enhanced soil exploration. This study also identified a QTL for As content in BR (qRAs11-1), in which we identified *LOC_Os11g10510*, a gene that encodes an alcohol dehydrogenase named OsADH2. The *low-arsenic line 3* (*las3*) mutant, in which OsADH2 is not functional, presents reduced amounts of As in aerial tissues and grains [52]. The decrease in As content in *las3* is correlated with the downregulation of genes encoding root silicate transporters (*LOW SILICON 1* and *2*), which are also known to play major roles in As uptake [70]. It has been proposed that under anaerobic conditions, the loss of function of OsADH2 leads to a decrease in ethanol fermentation, inducing a subsequent decrease in the cytosolic pH of root cells, which represses *LSI1* and *LSI2* expression and As uptake [52]. It is therefore likely that *OsADH2* and its alleles are responsive to the effect of qRA11-1 on As grain accumulation.

Among the new QTLs identified in this study, we identified several interesting candidate genes whose putative functions are related to ion uptake or homeostasis, and which could consequently affect the grain ion content. In qRAs10-1, for example, *LOC_Os10g30610* is upregulated in response to As [50]. This gene encodes an ABC transporter involved in exodermis suberization [49], that act as a barrier restricting ion diffusion such as Fe within the root cortex [71]. In fact, the overexpression of *LOC_Os10g30610* increased the extent of Casparian strips at the exodermis [71] and may contribute to reduce the inflow of As within the root. Therefore, the putative function of qRAs10-1 in As exclusion and qRAs11-1 in reduction of As transport may have a synergistic effect on lowering As uptake in rice and accumulation in grain, if combined. Furthermore, *LOC_Os05g04330* in qRP5-1 is involved in CHH DNA methylation maintenance, and the regulation of its expression in roots is dependent on the major phosphate starvation tolerance *Pup1* QTL [47]. These findings suggest that the response of rice to phosphate deficiency involves the regulation of gene expression via the modulation of DNA methylation. The involvement of these candidate genes in the regulation of grain ion accumulation needs further validation through functional analysis.

In conclusion, our association study on ion contents in BR and WR grains from a Vietnamese rice landrace collection confirmed several previously identified QTLs, provided new insights into the physiological mechanisms of some of these QTLs and identified new QTLs containing interesting potential candidate genes. Vietnamese landraces carrying QTLs for high Fe and Zn grain contents may represent new potential donors for rice grain biofortification, as these contents exceed the values targeted by breeding programs. Moreover, other donors carrying QTLs with complementary effects may be pyramided to reduce the As ion content in grain. Overall, the identification of these QTLs is valuable for improving the mineral ion content of rice grain while reducing the accumulation of toxic heavy metals.

## Supporting information

**S1 File. Inclusivity in global research.**
(DOCX)

**S1 Fig. Field trial planting in Hai Phong in 2019.**
(TIF)

**S2 Fig. Box plot representing variation in ion content in brown rice (BR) and white rice (WR) in the Vietnamese rice collection.** Ion content is represented as ppm. p-values from ANOVA tests are represented.
(TIF)

**S3 Fig. Localization of QTLs associated with ion content in Vietnamese rice.** The 12 rice chromosomes are represented. The vertical scale indicates the size of chromosomes in mega base pairs (Mb).
(TIF)

**S4 Fig. Genome-wide association mapping of K content in brown rice grain measured in the Vietnamese rice collection. A**. Manhattan plot representing the p-values of associations between SNPs and grain K content using the BLINK model in GAPIT. The solid green line represents the Bonferroni threshold and the dotted green line represents FDR threshold. **B**. Focus on the association between SNP Dj05_03417694F and grain K content on chromosome 5. **C**. Definition of QTL qRK5-1 based on blocks of linkage disequilibrium around Dj05_03417694F. The genes present in the QTL region are represented by grey rectangles. The selected candidate gene *LOC_Os05g06660* is highlighted in red. **D**. Box plots representing the

polymorphism distribution at SNP Dj05_03417694F in the Vietnamese collection. The number of genotypes in each haplotype group is represented. **E**. Expression levels of *LOC_Os05g06660* at different developmental stages. Expression data in different plant tissues were retrieved from the Rice Xpro database [44] as follow; LB_V: Leaf blade at vegetative stage; LB_Rp: Leaf blade at reproductive stage; LB_R: Leaf blade at ripening stage; LS_V: Leaf sheath at vegetative stage; LS_Rp: Leaf sheath at reproductive stage; R_V: Root at vegetative stage; R_Rp: Root at reproductive stage; S_Rp: Stem at reproductive stage; S_R: Stem at ripening stage; I: Inflorescence of 3.0–4.0 mm; A: Anther of 0.7–1.0 mm; Pis: Pistil of 10–14 cm panicle; L: Lemma of 4.0–5.0 mm floret; Pal: Palea of 4.0–5.0 mm floret; O: Ovary 3 days after flowering; Emb: Embryo at 14 days after flowering; End: Endosperm at 14 days after flowering. (TIF)

**S5 Fig. Genome-wide association mapping of As content in brown rice grain measured in the Vietnamese rice collection. A**. Manhattan plot representing the p-values of associations between SNPs and grain As content using the BLINK model in GAPIT. The solid green line represents the Bonferroni threshold and the dotted green line represents FDR threshold. **B**. Focus on the association between SNP Dj11_05821648F and grain As content on chromosome 11. **C**. Definition of QTL qRAs11-1 based on blocks of linkage disequilibrium around Dj11_05821648F. The genes present in the QTL region are represented by grey rectangles. The selected candidate gene *LOC_Os11g10510* is highlighted in red. **D**. Box plots representing the polymorphism distribution at SNP Dj11_05821648F in the Vietnamese collection. The number of genotypes in each haplotype group is represented. **E**. Expression levels of *LOC_Os11g10510* at different developmental stages. Expression data in different plant tissues were retrieved from the Rice Xpro database [44] as follow; LB_V: Leaf blade at vegetative stage; LB_Rp: Leaf blade at reproductive stage; LB_R: Leaf blade at ripening stage; LS_V: Leaf sheath at vegetative stage; LS_Rp: Leaf sheath at reproductive stage; R_V: Root at vegetative stage; R_Rp: Root at reproductive stage; S_Rp: Stem at reproductive stage; S_R: Stem at ripening stage; I: Inflorescence of 3.0–4.0 mm; A: Anther of 0.7–1.0 mm; Pis: Pistil of 10–14 cm panicle; L: Lemma of 4.0–5.0 mm floret; Pal: Palea of 4.0–5.0 mm floret; O: Ovary 3 days after flowering; Emb: Embryo at 14 days after flowering; End: Endosperm at 14 days after flowering. (TIF)

**S1 Table. Passport data of the different landraces present in the Vietnamese rice collection.** I: indica; J: japonica; m: admix; na: no data available. (XLSX)

**S2 Table. List of genes present in each QTL identified foreign content in the Vietnamese rice collection.** (XLSX)

## Acknowledgments

The authors would like to thank Nguyen Van Toan, Vu Thi Huong and Vu Manh An for their valuable support in field care and rice grain sample harvesting.

## Author Contributions

**Conceptualization:** Giang Thi Hoang, Alexandre Grondin, Pascal Gantet.

**Data curation:** Hien Linh Tran, Giang Thi Hoang, Nhung Thi Phuong Phung, Ham Huy Le.

**Formal analysis:** Hien Linh Tran, Giang Thi Hoang, Alexandre Grondin, Pascal Gantet.

**Funding acquisition:** Giang Thi Hoang, Alexandre Grondin, Pascal Gantet.

**Investigation:** Hien Linh Tran, Giang Thi Hoang.

**Project administration:** Giang Thi Hoang, Pascal Gantet.

**Supervision:** Giang Thi Hoang, Alexandre Grondin, Pascal Gantet.

**Writing – original draft:** Hien Linh Tran, Giang Thi Hoang, Alexandre Grondin, Pascal Gantet.

**Writing – review & editing:** Hien Linh Tran, Giang Thi Hoang, Nhung Thi Phuong Phung, Ham Huy Le, Alexandre Grondin, Pascal Gantet.

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
