## [Decision Letter · Decision Letter 0]

27 Sep 2024

PONE-D-24-17942Quantitative trait loci for grain mineral element accumulation in Vietnamese rice landracesPLOS ONE

Dear Dr. Grondin,

Thank you for submitting your manuscript to PLOS ONE. After careful consideration, we feel that it has merit but does not fully meet PLOS ONE’s publication criteria as it currently stands. Therefore, we invite you to submit a revised version of the manuscript that addresses the points raised during the review process.

**ACADEMIC EDITOR:  **I recommend that the authors address the reviewers' comments and revise the manuscript accordingly.==============================

We look forward to receiving your revised manuscript.

Kind regards,

Karthikeyan Adhimoolam

Academic Editor

PLOS ONE

Journal Requirements:

2. Please remove all personal information, ensure that the data shared are in accordance with participant consent, and re-upload a fully anonymized data set. 

3. Please include a complete copy of PLOS’ questionnaire on inclusivity in global research in your revised manuscript. Our policy for research in this area aims to improve transparency in the reporting of research performed outside of researchers’ own country or community. The policy applies to researchers who have travelled to a different country to conduct research, research with Indigenous populations or their lands, and research on cultural artefacts. The questionnaire can also be requested at the journal’s discretion for any other submissions, even if these conditions are not met.  Please find more information on the policy and a link to download a blank copy of the questionnaire here: https://journals.plos.org/plosone/s/best-practices-in-research-reporting. Please upload a completed version of your questionnaire as Supporting Information when you resubmit your manuscript.

4. In your Methods section, please provide additional information regarding the permits you obtained for the work. Please ensure you have included the full name of the authority that approved the field site access and, if no permits were required, a brief statement explaining why.

5. Thank you for stating the following financial disclosure: Hien Linh Tran is supported by the France Excellence Scholarship Program from the French Embassy in Vietnam. This work was supported by the “QUARION” EPPN European Plant Phenotyping Network (EPPN) 2020 project, the “RICE-VN” TRENPLIN-ASEAN prize awarded by the French Academy of Science and the French Ministry of Higher Education and Research (2023-2026). 

6. Thank you for stating the following in the Acknowledgments Section of your manuscript: Hien Linh Tran is supported by the France Excellence Scholarship Program from the French Embassy in Vietnam. This work was supported by the “QUARION” EPPN European Plant Phenotyping Network (EPPN) 2020 project, the “RICE-VN” TRENPLIN-ASEAN prize awarded by the French Academy of Science and the French Ministry of Higher Education and Research (2023-2026). The authors would like to thank Nguyen Van Toan, Vu Thi Huong and Vu Manh An for their valuable support in field care and rice grain sample harvesting.

Please remove any funding-related text from the manuscript and let us know how you would like to update your Funding Statement. Currently, your Funding Statement reads as follows: Hien Linh Tran is supported by the France Excellence Scholarship Program from the French Embassy in Vietnam. This work was supported by the “QUARION” EPPN European Plant Phenotyping Network (EPPN) 2020 project, the “RICE-VN” TRENPLIN-ASEAN prize awarded by the French Academy of Science and the French Ministry of Higher Education and Research (2023-2026). 

7. When completing the data availability statement of the submission form, you indicated that you will make your data available on acceptance. We strongly recommend all authors decide on a data sharing plan before acceptance, as the process can be lengthy and hold up publication timelines. Please note that, though access restrictions are acceptable now, your entire data will need to be made freely accessible if your manuscript is accepted for publication. This policy applies to all data except where public deposition would breach compliance with the protocol approved by your research ethics board. If you are unable to adhere to our open data policy, please kindly revise your statement to explain your reasoning and we will seek the editor's input on an exemption. Please be assured that, once you have provided your new statement, the assessment of your exemption will not hold up the peer review process.

Reviewers' comments:

Reviewer's Responses to Questions

**Comments to the Author**

1. Is the manuscript technically sound, and do the data support the conclusions?

Reviewer #1: Yes

Reviewer #2: Yes

Reviewer #3: Yes

2. Has the statistical analysis been performed appropriately and rigorously? 

Reviewer #1: Yes

Reviewer #2: Yes

Reviewer #3: Yes

3. Have the authors made all data underlying the findings in their manuscript fully available?

Reviewer #1: No

Reviewer #2: Yes

Reviewer #3: Yes

4. Is the manuscript presented in an intelligible fashion and written in standard English?

Reviewer #1: No

Reviewer #2: Yes

Reviewer #3: Yes

5. Review Comments to the Author

Reviewer #1: Quantitative trait loci for grain mineral element accumulation in Vietnamese rice landraces.

# Title: Rewriting suggested.

Abstract: OK

Keywords: OK

A dedicated Abbreviation section suggested (including transporters, QTLs etc.).

Authors are advised to add a note on deprivation of elements among the population (age and sex wise etc.).

Role of As and Cd in minimizing elemental content in raw and cooked rice.

Clearly mention the gap area and aim and scope of the study (bullet).

Line 133: 50 cm spacing or 5.0 cm spacing?

SRM recovery should be presented in tabular format in supplementary file.

Line:168: mean of how many replications?

In the discussion section, authors are required to interpret the varieties having greater elemental profile linked with QTL in terms of RDA or RDI for greater acceptability.

Moreover, how climatic fluctuations can influence the elemental profile will be highly appreciated.

Reviewer #2: The manuscript entitled “Quantitative trait loci for grain mineral element accumulation in Vietnamese rice landraces” focuses on identifying the QTL regions and candidate genes for different mineral elements in rice landraces. The manuscript is clear, concise, and informative. The methods are appropriate and the results are presented in a logical and transparent manner. The discussion is balanced and comprehensive. I have only a few minor comments and suggestions for improvement:

Introduction: The Introduction has been well drafted, with extensive collection of points related to the study, highlighting the rationale behind it. However, the chronology of these sentences may have to be organised for clarity. The following corrections may be carried out for improvement of the manuscript,

Comment 1: Line 42–62, 1st paragraph contains the brief introduction about biofortification and its improvement, while 2nd paragraph (line 64-81) contains other elements and its correlation with harmful heavy metals, followed by its improvement. This could be modified as stating brief introduction of different elements in 1st paragraph and improvement strategies in 2nd paragraph.

Comment 2: Line 83, introductory sentences need to be added to this paragraph. Instead of starting with different GWAS studies, mention about other mapping strategies and how advantageous GWAS is over them with respect to grain ions trait.

Comment 3: Line 108, the concluding statement, which defines the objective of the study needs clarity. Just mention the objective and delete the findings, which may be discussed in results section.

Materials and methods: The methodology has been presented well in different subsections viz., Plant materials and genotypic data, Field experiment, Grain ion phenotyping, Statistical analysis, Genome-wide association studies and Linkage disequilibrium (LD) heatmap and identification of candidate genes. However, the following corrections may be carried out in the manuscript for its improvement,

Comment 4: Were the ions estimated in soil where the crop has been raised? And mention crop production methods, whether any nutrients were supplemented?

Comment 5: Line 168, “Means, standard deviations, and coefficients of variation” may be modified as “Mean, standard deviation, and coefficients of variation”

Comment 6: Line 175, mention the model which has been used for calculation of BLUEs for your incomplete block design.

Results: The results has been presented well in different subsections viz., Variation in grain ion content in Vietnamese rice landraces, Identification of QTL related to grain ion content in Vietnamese rice landraces and Potential candidate genes related to grain ion content in Vietnamese rice landraces. However, the following corrections may be carried out in the manuscript for its improvement,

Comment 7: Line 220, “analyses were performed” may be modified as “analyses was performed”

Comment 8: Line 227, it is “(R < -0.34)” or “(R = -0.34)”? check all instances.

Comment 8: Line 228, it is “co-variations” or “correlations”?

Comment 9: Table 2 may be modified for clarity. It should be organised based on the traits (ions). Additionally, points in the results section should reflect in the table (model in which it was associated, bonferroni threshold, etc.).

Comment 10: Line 247-258, these results may also be added for investigation of candidate genes in Table 3. The relevance for these genes with associated traits should be established, or else delete it.

Comment 11: Line 266, is it 3.6% or 3.6ppm? check all instances

Comment 12: Line 261-287, the paragraphs looks more like a discussion. Modify it. The results section for these should only contain the findings of this study.

Discussion: The discussion has been presented well and schematically organised. However, the following corrections may be carried out in the manuscript for its improvement,

Comment 13: Line 311, modify “with K, Mg, P showing smallest variation and Fe, Zn” as “with K, Mg and P showing smallest variation, while, Fe, Zn”

Abstract: The abstract is well drafted and sequentially written which reflects the contents of the manuscript. However, the corrections if made in the above sections need to be addressed in abstract also.

General comments:

1. Check whether the reference format is according to the journal style

2. Grammatical and typographic errors may be checked and corrected

Reviewer #3: Overall the findings of the study is compelling. Although there are few typographical error in some statements that could also be addressed, the efforts made by the authors are appreciable. Moreover, the findings of this study is almost agreed to support the previous works, QTL mapping studies for instance.

However, I have listed few corrections for the authors to look over it.

•Correct the typographical and grammatical errors in the following line numbers, 98,112,119,155 169 etc…

•Did authors measured phytic acid levels? If so it would be good to included phytic acid contents in WR and BR grains

•Check for the references as per the journal format!

6. PLOS authors have the option to publish the peer review history of their article (what does this mean?). If published, this will include your full peer review and any attached files.

Reviewer #1: **Yes: **Dr. Debojyoti Moulick

Reviewer #2: **Yes: **Manoharan Akilan

Reviewer #3: **Yes: **Vellaichamy Gandhimeyyan Renganathan

---

## [Author Response · Author response to Decision Letter 0]

30 Oct 2024

Reviewer #1: Quantitative trait loci for grain mineral element accumulation in Vietnamese rice landraces.

Author’s response: We thank the Reviewer for the constructive comments on the manuscript.

# Title: Rewriting suggested.

Abstract: OK

Keywords: OK

A dedicated Abbreviation section suggested (including transporters, QTLs etc.).

Author’s response: We have added an Abbreviation section and tried to reduce the abbreviations within the text. Gene/protein abbreviations are consistently preceded by their full name. Abbreviations for these were not included because they do not appear throughout the text but rather in specific sections. 

Authors are advised to add a note on deprivation of elements among the population (age and sex wise etc.).

Author’s response: We now refer in the first paragraph to Fe and Zn deprivation repartition in the world and population.

Role of As and Cd in minimizing elemental content in raw and cooked rice.

Author’s response: We thank the Reviewer for this comment. We have now added a description of these aspects in the first paragraph of the introduction.

Clearly mention the gap area and aim and scope of the study (bullet).

Author’s response: We have clarified the gap, aim and scope of the study according to the Reviewer’s comment.

Line 133: 50 cm spacing or 5.0 cm spacing?

Author’s response: In the field experiment, we observed 25 cm spacing between and within rows, 50 cm spacing between blocks and 75 cm spacing between flowering groups. We have now clarified that in the manuscript.

SRM recovery should be presented in tabular format in supplementary file.

Author’s response: The ionomic analysis did not include measurements on certified reference material. However, calibration standards from purchased certified standards were used to calibrate the instrument and “pooled sample” (a small amount taken from each of the samples analysed and combined) were analysed after every nine samples in all of the batches of analysis to ensure consistency between analysis batches as indicated in the material and methods. We acknowledge that certified reference material would have been useful to confirm absolute concentration values in our samples but our analysis rather ensured that every sample was treated similarly to allow comparison between genotypes from the panel. Furthermore, we observed values within the order of magnitude of the ion contents usually observed in rice. Therefore, we do not expect major bias in the absolute values of ion content in rice grains we measured in our analysis. 

Line:168: mean of how many replications?

Author’s response: It is the mean of three replications. We have added this information in the text. 

In the discussion section, authors are required to interpret the varieties having greater elemental profile linked with QTL in terms of RDA or RDI for greater acceptability.

Moreover, how climatic fluctuations can influence the elemental profile will be highly appreciated.

Author’s response: We have mentioned the international breeding targets set for rice Fe and Zn content in the discussion and we added a sentence on the importance of developing climate resilient rice varieties while ensuring grain quality. 

Reviewer #2: The manuscript entitled “Quantitative trait loci for grain mineral element accumulation in Vietnamese rice landraces” focuses on identifying the QTL regions and candidate genes for different mineral elements in rice landraces. The manuscript is clear, concise, and informative. The methods are appropriate and the results are presented in a logical and transparent manner. The discussion is balanced and comprehensive. 

Author’s response: We thank the Reviewer for the constructive comments on the manuscript.

I have only a few minor comments and suggestions for improvement:

Introduction: The Introduction has been well drafted, with extensive collection of points related to the study, highlighting the rationale behind it. However, the chronology of these sentences may have to be organised for clarity. The following corrections may be carried out for improvement of the manuscript,

Comment 1: Line 42–62, 1st paragraph contains the brief introduction about biofortification and its improvement, while 2nd paragraph (line 64-81) contains other elements and its correlation with harmful heavy metals, followed by its improvement. This could be modified as stating brief introduction of different elements in 1st paragraph and improvement strategies in 2nd paragraph.

Author’s response: We have re-organised the two paragraphs according to the Reviewer’s comment.

Comment 2: Line 83, introductory sentences need to be added to this paragraph. Instead of starting with different GWAS studies, mention about other mapping strategies and how advantageous GWAS is over them with respect to grain ions trait.

Author’s response: We added a sentence in the introduction describing the advantages of GWAS over multiparental populations in detecting QTLs associated with ion accumulation. 

Comment 3: Line 108, the concluding statement, which defines the objective of the study needs clarity. Just mention the objective and delete the findings, which may be discussed in results section.

Author’s response: We have clarified the objectives and removed the findings.

Materials and methods: The methodology has been presented well in different subsections viz., Plant materials and genotypic data, Field experiment, Grain ion phenotyping, Statistical analysis, Genome-wide association studies and Linkage disequilibrium (LD) heatmap and identification of candidate genes. However, the following corrections may be carried out in the manuscript for its improvement,

Comment 4: Were the ions estimated in soil where the crop has been raised? And mention crop production methods, whether any nutrients were supplemented?

Author’s response: We did not measure ion content in the soil of the field where plants were grown, but, as far as we know, no ion deficiency/toxicity in the soil of the region nearby was reported. The field experiment was fertilized as per the usual farmer-based fertilization recommended in the region. In fact, the quantities of ions that we observed in the grains are within the order of magnitude of the ion contents usually observed in rice as mentioned in the article. Therefore, we do not suspect that a bias related to the abnormal presence of an ion in the soil has occurred.

Comment 5: Line 168, “Means, standard deviations, and coefficients of variation” may be modified as “Mean, standard deviation, and coefficients of variation”

Author’s response: We have edited the text accordingly.

Comment 6: Line 175, mention the model which has been used for calculation of BLUEs for your incomplete block design.

Author’s response: We have clarified the sentence indicating the R package used (StatgenSTA), the function within the package (SpATS) and the equation describing the model.

Results: The results has been presented well in different subsections viz., Variation in grain ion content in Vietnamese rice landraces, Identification of QTL related to grain ion content in Vietnamese rice landraces and Potential candidate genes related to grain ion content in Vietnamese rice landraces. However, the following corrections may be carried out in the manuscript for its improvement,

Comment 7: Line 220, “analyses were performed” may be modified as “analyses was performed”

Author’s response: We have edited the text accordingly.

Comment 8: Line 227, it is “(R < -0.34)” or “(R = -0.34)”? check all instances.

Author’s response: We have changed the text and now indicate the coefficient of correlation for both brown rice and white rice.

Comment 8: Line 228, it is “co-variations” or “correlations”?

Author’s response: We have now used the term “associations” to indicate that ion content in brown rice may be associated with ion content in white rice as illustrated by the correlations for P, K and Mg between the two grain types.

Comment 9: Table 2 may be modified for clarity. It should be organised based on the traits (ions). Additionally, points in the results section should reflect in the table (model in which it was associated, bonferroni threshold, etc.).

Author’s response: We have modified Table 2 according to the Reviewer’s comment.

Comment 10: Line 247-258, these results may also be added for investigation of candidate genes in Table 3. The relevance for these genes with associated traits should be established, or else delete it.

Author’s response: These sentences refer to colocalisation between the QTL associated with ion content found in this study and QTL associated with other traits in the same rice collection. None of these former QTL correspond to those in which we have identified interesting candidate genes. Therefore, the colocalization information cannot be added to Table 3. In Table 3 we have removed the LOC_Os04g42930 gene in qRMn4-1 as we acknowledge its function was not clearly associated with Mn acquisition and translocation. 

Comment 11: Line 266, is it 3.6% or 3.6ppm? check all instances

Author’s response: We confirm that the T allele induces an increase in grain P content of 3.6% on average. We have changed the sentence to “3.6% increase in average” in qRP5-1. We have also changed the sentence describing the reduction of 24% in average for As content provided by the C allele in qRAs-10-1. 

Comment 12: Line 261-287, the paragraphs looks more like a discussion. Modify it. The results section for these should only contain the findings of this study.

Author’s response: We appreciate the Reviewer’s feedback. However, we support the selection of these genes by their presence within the QTL interval which we determined, and their expression data or functional information which has been retrieved from databases or the references we are citing in this section. As such, this section presents information that we believe are appropriately placed and necessary within the results section, particularly as we were careful not to make this redundant with the discussion. This structure is also consistent with our previous publications (Debieu et al., 2018 and Affortit et al., 2022), and for these reasons, we have decided to retain these information in the results rather than moving it to the discussion. 

Discussion: The discussion has been presented well and schematically organised. However, the following corrections may be carried out in the manuscript for its improvement,

Comment 13: Line 311, modify “with K, Mg, P showing smallest variation and Fe, Zn” as “with K, Mg and P showing smallest variation, while, Fe, Zn”

Author’s response: We have changed the text according to the Reviewers comment.

Abstract: The abstract is well drafted and sequentially written which reflects the contents of the manuscript. However, the corrections if made in the above sections need to be addressed in abstract also.

Author’s response: We have edited the abstract accordingly.

General comments:

1. Check whether the reference format is according to the journal style

Author’s response: We have revised the reference to align with the journal’s style.

2. Grammatical and typographic errors may be checked and corrected

Author’s response: We have diligently corrected these errors throughout the text. In addition, the paper underwent English editing by a private company. 

Reviewer #3: Overall the findings of the study is compelling. Although there are few typographical error in some statements that could also be addressed, the efforts made by the authors are appreciable. Moreover, the findings of this study is almost agreed to support the previous works, QTL mapping studies for instance.

Author’s response: We thank the Reviewer for the constructive comments on the manuscript.

However, I have listed few corrections for the authors to look over it.

•Correct the typographical and grammatical errors in the following line numbers, 98,112,119,155 169 etc…

Author’s response: We have diligently corrected these errors throughout the text. In addition, the paper underwent English editing by a private company.

•Did authors measured phytic acid levels? If so it would be good to included phytic acid contents in WR and BR grains

Author’s response: Indeed, it would have been interesting to measure phytic acid levels as it represents a strong chelator of metal cations and reduces the bioavailability of important micronutrients. However, due to budget limitations phytic acid were not measured in our experiments.

•Check for the references as per the journal format!

Author’s response: We have revised the reference to align with the journal’s style.

---

## [Decision Letter · Decision Letter 1]

20 Nov 2024

PONE-D-24-17942R1Quantitative trait loci for grain mineral element accumulation in Vietnamese rice landracesPLOS ONE

Dear Dr. Grondin,

Thank you for submitting your manuscript to PLOS ONE. After careful consideration, we feel that it has merit but does not fully meet PLOS ONE’s publication criteria as it currently stands. Therefore, we invite you to submit a revised version of the manuscript that addresses the points raised during the review process.

We look forward to receiving your revised manuscript.

Kind regards,

Karthikeyan Adhimoolam

Academic Editor

PLOS ONE

Journal Requirements:

Additional Editor Comments:

**Minor revision**

Reviewers' comments:

Reviewer's Responses to Questions

**Comments to the Author**

1. If the authors have adequately addressed your comments raised in a previous round of review and you feel that this manuscript is now acceptable for publication, you may indicate that here to bypass the “Comments to the Author” section, enter your conflict of interest statement in the “Confidential to Editor” section, and submit your "Accept" recommendation.

Reviewer #1: All comments have been addressed

Reviewer #2: All comments have been addressed

2. Is the manuscript technically sound, and do the data support the conclusions?

Reviewer #1: Yes

Reviewer #2: Yes

3. Has the statistical analysis been performed appropriately and rigorously? 

Reviewer #1: Yes

Reviewer #2: Yes

4. Have the authors made all data underlying the findings in their manuscript fully available?

Reviewer #1: Yes

Reviewer #2: Yes

5. Is the manuscript presented in an intelligible fashion and written in standard English?

Reviewer #1: No

Reviewer #2: Yes

6. Review Comments to the Author

Reviewer #1: It seems the authors have answered the queries raised by the reviewers in a more or less satisfactory manner and improvement in quality of MS in R1 version can be seen.However, the final call will be from editor's desk.

Reviewer #2: Reviewer’s comments for the author’s response

I have carefully reviewed the revised manuscript and the author’s responses to my comments. I am pleased to report that the authors have adequately addressed my suggestions and made the necessary revisions to improve the clarity and quality of the manuscript. I would suggest one more change to be made to improve the quality of the manuscript.

Comment 1: Table 2, differentiate the SNPs that are associated in multiple GWAS models (14 SNPs) and which are exclusively associated in BLINK model (13 SNPs) using footnotes.

7. PLOS authors have the option to publish the peer review history of their article (what does this mean?). If published, this will include your full peer review and any attached files.

Reviewer #1: **Yes: **Dr. Debojyoti Moulick

Reviewer #2: **Yes: **Manoharan Akilan

---

## [Author Response · Author response to Decision Letter 1]

22 Nov 2024

Reviewer #1: It seems the authors have answered the queries raised by the reviewers in a more or less satisfactory manner and improvement in quality of MS in R1 version can be seen. However, the final call will be from editor's desk.

Author’s response: We thank the Reviewer for the comments on the manuscript.

Is the manuscript presented in an intelligible fashion and written in standard English? Reviewer #1: No

Author’s response: As stated in our previous round of revisions, the manuscript have undergone English language editing by the American Journal Experts company, and the editing certificate is included with the submission. We have further refined the clarity of the manuscript and made minor edits in the text. A tracked changes version is provided as part of the resubmission. We believe that these revisions have further improved the clarity of the manuscript. However, we remain open to addressing further clarification the reviewers may advise.

Reviewer #2: I have carefully reviewed the revised manuscript and the author’s responses to my comments. I am pleased to report that the authors have adequately addressed my suggestions and made the necessary revisions to improve the clarity and quality of the manuscript. I would suggest one more change to be made to improve the quality of the manuscript.

Author’s response: We thank the Reviewer for the comments on the manuscript.

Comment 1: Table 2, differentiate the SNPs that are associated in multiple GWAS models (14 SNPs) and which are exclusively associated in BLINK model (13 SNPs) using footnotes.

Author’s response: We have revised Table 2 in accordance with Reviewer 2’s suggestions.

---

## [Decision Letter · Decision Letter 2]

29 Nov 2024

Quantitative trait loci for grain mineral element accumulation in Vietnamese rice landraces

PONE-D-24-17942R2

Dear Dr. Grondin,

We’re pleased to inform you that your manuscript has been judged scientifically suitable for publication and will be formally accepted for publication once it meets all outstanding technical requirements.

Kind regards,

Karthikeyan Adhimoolam

Academic Editor

PLOS ONE

Additional Editor Comments (optional):

Reviewers' comments:

Reviewer's Responses to Questions

**Comments to the Author**

1. If the authors have adequately addressed your comments raised in a previous round of review and you feel that this manuscript is now acceptable for publication, you may indicate that here to bypass the “Comments to the Author” section, enter your conflict of interest statement in the “Confidential to Editor” section, and submit your "Accept" recommendation.

Reviewer #2: All comments have been addressed

2. Is the manuscript technically sound, and do the data support the conclusions?

Reviewer #2: Yes

3. Has the statistical analysis been performed appropriately and rigorously? 

Reviewer #2: Yes

4. Have the authors made all data underlying the findings in their manuscript fully available?

Reviewer #2: Yes

5. Is the manuscript presented in an intelligible fashion and written in standard English?

Reviewer #2: Yes

6. Review Comments to the Author

Reviewer #2: Reviewer’s comments for the author’s response

I have carefully reviewed the revised manuscript “Quantitative trait loci for grain mineral element accumulation in Vietnamese rice landraces” and the author’s responses to my comments. I am pleased to report that the authors have adequately addressed all of my suggestions and made the necessary revisions to improve the clarity and quality of the manuscript. The authors have demonstrated a thorough consideration of the feedback, and the manuscript now meets the standards required for publication. I find no further revisions necessary and would recommend accepting the manuscript for publication.

7. PLOS authors have the option to publish the peer review history of their article (what does this mean?). If published, this will include your full peer review and any attached files.

Reviewer #2: **Yes: **Manoharan Akilan

---

## [Editor Report · Acceptance letter]

11 Dec 2024

PONE-D-24-17942R2 

PLOS ONE

Dear Dr. Grondin, 

I'm pleased to inform you that your manuscript has been deemed suitable for publication in PLOS ONE. Congratulations! Your manuscript is now being handed over to our production team.

Kind regards, 

on behalf of

Dr. Karthikeyan Adhimoolam 

Academic Editor

PLOS ONE